# Revisiting Representation Learning for Singing Voice Separation with Sinkhorn Distances

**Stylianos I. Mimilakis** [* 1]   **Konstantinos Drossos** [* 2]   **Gerald Schuller** [3]

## Abstract

In this work we present a method for unsupervised learning of audio representations, focused on the task of singing voice separation. We build upon a previously proposed method for learning representations of time-domain music signals with a re-parameterized denoising autoencoder, extending it by using the family of Sinkhorn distances with entropic regularization. We evaluate our method on the freely available MUSDB18 dataset of professionally produced music recordings, and our results show that Sinkhorn distances with small strength of entropic regularization are marginally improving the performance of informed singing voice separation. By increasing the strength of the entropic regularization, the learned representations of the mixture signal consists of almost perfectly additive and distinctly structured sources.

## 1. Introduction

Music source separation aims at the estimation of the individual music sources of an observed mixture signal. To that aim, supervised deep learning (DL) based approaches are shown to yield remarkable results (Hennequin et al., 2019; Défossez et al., 2019; Stöter et al., 2019; Samuel et al., 2020). Although different types of sources can be estimated from a music mixture, a specific task of music source separation that has received a lot of attention in relevant research communities is the separation of the singing voice, or singing voice source separation (Rafii et al., 2018).

State-of-the-art approaches in DL-based music and singing voice source separation have considered using both pre-computed and learned signal representations. The approaches that utilize pre-computed signal representations, have extensively employed the short-time Fourier transform (STFT) (Hennequin et al., 2019; Stöter et al., 2019; Drossos et al., 2018; Mimilakis et al., 2018). On the other hand, learned representations are commonly used in end-to-end models and are jointly learned with the parameters of the rest of the model.

In both of the previous approaches, the learning of the representations is based on objectives that assess the reconstruction of the signals of the target sources (Défossez et al., 2019; Samuel et al., 2020). In many cases, the approaches based on end-to-end models do not yield better performance than approaches using representations computed using the STFT (Défossez et al., 2019; Samuel et al., 2020; Tzinis et al., 2020). Furthermore, the learned representations obtained by approaches utilizing end-to-end models are not easily nor intuitively interpreted, compared to the typical STFT representation that utilizes pre-computed signal representations. In order to bridge the gap of separation performance and interpretability between end-to-end-based and STFT-based approaches, recent studies focus on representation learning (Tzinis et al., 2020; Mimilakis et al., 2020).

In (Tzinis et al., 2020) is presented a sound source separation method, focused on representation learning. An encoder gets as an input the signals of the sources and their corresponding mixture, and outputs latent representations of the signals of each source and the mixture. Then, using these latent representations, the method calculates and applies source dependent masks to the latent representation of mixture. The result of the application of masks is given as an input to the decoder, which outputs an estimation of the signal of each source. The encoder and the decoder are jointly optimized to minimize the reconstruction error between the ground truth and estimated signals of each source. However, using reconstruction objectives for the separation of only specific sources could severely restrict the representation learning capabilities of encoder-decoder methods (Vincent, 2011). In (Mimilakis et al., 2020) it is proposed to learn representations for singing voice separation in an unsupervised way using a re-parameterized denoising autoencoder (DAE) (Vincent et al., 2010). The re-parameterization replaces the *decoding* basis functions by amplitude-modulated cosine functions whose parameters are learned with the rest

---

[*]Equal contribution   [1]Fraunhofer-IDMT, Ilmenau, Germany   [2]Audio Research Group, Tampere University, Tampere, Finland   [3]Applied Media Systems Group, Technical University of Ilmenau, Ilmenau, Germany. Correspondence to: Stylianos I. Mimilakis <mis@idmt.fraunhofer.de>.

*Proceedings of the 37th International Conference on Machine Learning*, Vienna, Austria, PMLR 108, 2020. Copyright 2020 by the author(s).

of the DAE. This results into an interpretable representation of the singing voice signal that conveys amplitude information for modulated sinusoidal bases. The re-parametization is similar to Sinc-Networks (Ravanelli & Bengio, 2018) that use sinc functions for *encoding* speech signals. The parameters of the denoising autoencoder employed in (Mimilakis et al., 2020) are optimized using two objectives. The first objective is to minimize the reconstruction error between the clean and the reconstructed signal voice signal, and the second objective enforces the smoothness of the mixture signal's representation.

In this work we focus on unsupervised representation learning and we aim at learning representations of music signals that can offer enhanced interpretability combined with improved source separation performance. We build on the work presented in (Mimilakis et al., 2020) and we extend it by using the Sinkhorn distances with entropic regularization (Cuturi, 2013) as a representation specific objective. Our contribution is to experimentally show that Sinkhorn distances with entropic regularization can assist in learning representations in which the sources can be efficiently separated and the representations of sources are distinctly structured and additive.

### Notation

Bold lowercase letters, e.g., "$\mathbf{x}$", denote vectors and bold uppercase letters, e.g. "$\mathbf{X}$", denote matrices. The $l$-th element of a vector is denoted as $x_{[l]}$. Similarly, accessing elements from matrices is denoted as $X_{[l,l']}$.

## 2. Proposed method

Our method follows the one presented in (Mimilakis et al., 2020) and employs an encoder $E(\cdot)$ and a decoder $D(\cdot)$. The input to our method is a music signal, $\mathbf{x} \in \mathbb{R}^N$, with $N$ time-domain samples. The output of the method is the learned non-negative representation of $\mathbf{x}$, $\mathbf{A} \in \mathbb{R}_{\geq 0}^{C \times T}$, with $T$ templates of $C$ features. The $C$ features can be viewed as analogous to the frequency bins and the $T$ templates as the analogous to the time-frames in a time-frequency representation. $\mathbf{A}$ is computed by the encoder $E(\cdot)$, and is interpreted as the magnitude information for a real-valued, sinusoidal-based model, employed by the decoder $D(\cdot)$.

To optimize $E(\cdot)$, we employ the decoder $D(\cdot)$ and a dataset of monaural (single channel) recordings of singing voice, $\mathbf{x}_v \in \mathbb{R}^N$, and accompanying musical instruments. Using $\mathbf{x}_v$ we create two synthetic signals. The first synthetic signal, $\tilde{\mathbf{x}}_m \in \mathbb{R}^N$, is the result of an additive corruption process, where the accompanying musical instruments such as drums, guitars, synthesizers, and bass (i.e. a generic multi-modal distribution-based noise) are added to $\mathbf{x}_v$. The second synthetic signal, $\tilde{\mathbf{x}}_v \in \mathbb{R}^N$, is also the result of a corruption process, where Gaussian noise is added to $\mathbf{x}_v$, independently of the amplitude of $\mathbf{x}_v$.

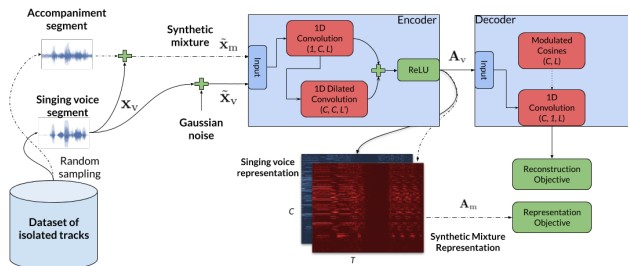

*Figure 1.* Overview of our proposed method for representation learning.

During the optimization process (i.e. training), the encoder $E(\cdot)$ computes two non-negative representations $\mathbf{A}_m$, $\mathbf{A}_v \in \mathbb{R}_{\geq 0}^{C \times T}$ using the two above mentioned synthetic signals, $\tilde{\mathbf{x}}_m$ and $\tilde{\mathbf{x}}_v$, respectively. $\mathbf{A}_v$ is used as input to $D(\cdot)$, and $D(\cdot)$ outputs an approximation of the clean singing voice signal $\mathbf{x}_v$, $\hat{\mathbf{x}}_v$. $\mathbf{A}_m$ is solely used to calculate an extra loss that will allow $E(\cdot)$ to learn information regarding the additive multi-modal noise (Mimilakis et al., 2020). An illustration of the training procedure in Figure 1. After the optimization process, $E(\cdot)$ can take as an input any musical signal $\mathbf{x}$, and will output the representation of $\mathbf{x}$, $\mathbf{A}$. The benefit is that $\mathbf{A}$ has good interpretability attributes, e.g. is non-negative, has structured spectrogram representation, and can be effectively used in the downstream task of singing voice separation.

### 2.1. Encoder

The encoder $E(\cdot)$ consists of two one-dimensional (1D) convolutions with strides. The first 1D convolution uses a stride $S$ and a set of $C$ number of kernels, $\mathbf{k}_c \in \mathbb{R}^L$ where $L$ is the temporal length of each $\mathbf{k}$. The first convolution takes as inputs the signals $\tilde{\mathbf{x}}_m$ and $\tilde{\mathbf{x}}_v$, and outputs the learned latent representations $\tilde{\mathbf{H}}_m \in \mathbb{R}_{\geq 0}^{C \times T}$ and $\tilde{\mathbf{H}}_v \in \mathbb{R}_{\geq 0}^{C \times T}$, respectively, using

$$\tilde{H}_{\star[c,t]} = \sum_{l=0}^{L-1} \tilde{x}_{\star[St+l]} k_{c[l]}, \qquad (1)$$

where "$\star$" refers to either "m" or "v" for brevity, and $t \in [0, \dots, T-1]$. Appropriate zero-padding is applied to $\tilde{\mathbf{x}}_\star$, so that $T = \lceil N/S \rceil$, where $\lceil \cdot \rceil$ is the ceiling function. Each $\tilde{\mathbf{H}}_\star$ is used as an input to the second 1D convolution, which uses another set of $C$ kernels, $\mathbf{K}'_{c'} \in \mathbb{R}^{L' \times C}$, where $c' = [1, \dots, C]$, with a temporal length $L'$ that is $L' << L$. The output of the second convolution is $\mathbf{H}_\star \in \mathbb{R}^{C \times T}$, and is performed with a dilation factor of $\phi$ and a unit stride, as

$$H_{\star[c',t]} = \sum_{c=0}^{C-1} \sum_{l'=0}^{L'-1} \tilde{H}_{\star[c,t+\phi l']} K'_{c'[l',c]}. \qquad (2)$$

Then, each $\mathbf{H}_\star$ is used in a residual connection, followed by the application of the rectified linear unit (ReLU) func-

tion (Nair & Hinton, 2010), as

$$\mathbf{A}_\star = \text{ReLU}(\mathbf{H}_\star + \tilde{\mathbf{H}}_\star). \qquad (3)$$

This is performed in order to enforce smooth and non-negative representations. The smoothness and the non-negativity are attributes that can enhance interpretability and are useful for the separation of audio and music sources (Smaragdis & Venkataramani, 2017). To further enforce the smooth representations under realistic corruption processes, in (Mimilakis et al., 2020) it is proposed to minimize the (anisotropic) total-variation denoising cost function, $\mathcal{L}_{\text{TV}}$ (Rudin et al., 1992), of the representation $\mathbf{A}_{\text{m}}$. $\mathcal{L}_{\text{TV}}$ is computed as

$$\mathcal{L}_{\text{TV}}(\mathbf{A}_{\text{m}}) = \frac{1}{CT}\Big(\sum_{c=1}^{C-1}\sum_{t=0}^{T-1}|\text{A}_{\text{m}[c,t]} - \text{A}_{\text{m}[c-1,t]}|$$
$$+ \sum_{t=1}^{T-1}\sum_{c=0}^{C-1}|\text{A}_{\text{m}[c,t]} - \text{A}_{\text{m}[c,t-1]}|\Big). \qquad (4)$$

Practically, $\mathcal{L}_{\text{TV}}$ penalizes $E(\cdot)$ by the norm of the first order difference across the time-frames $T$ and templates $C$, promoting slow time varying representations and grouping of the template activity. The previously mentioned representation attributes are formed from domain knowledge that is based on the STFT.

According to (Arjovsky et al., 2017)(Theorem 2) the total-variation distance, in our particular case the sum of absolute differences employed in Eq.(4), is not a suitable cost function for data distributions supported by low-dimensional manifolds. Instead, optimal transportation distances are suitable. We hypothesize that the singing voice, the mixture signals, and their corresponding representations can be described by low-dimensional manifolds, and we propose to replace $\mathcal{L}_{\text{TV}}$ by Sinkhorn distances, $\mathcal{L}_{\text{SK}}$. This is because $\mathcal{L}_{\text{SK}}$ allow an efficient computation of optimal transportation cost (Cuturi, 2013). More specifically, we use

$$\mathcal{L}_{\text{SK}}(\mathbf{A}_{\text{m}}) = \langle\mathbf{P}_\lambda, \psi(\mathbf{A}_{\text{m}})\rangle, \qquad (5)$$

where $\langle\cdot,\cdot\rangle$ is the Frobenious dot-product and $\psi : \mathbb{R}_{\geq 0}^{C\times T} \mapsto \mathbb{R}_{\geq 0}^{T\times T}$ is a function that computes the cost matrix $\mathbf{M} \in \mathbb{R}_{\geq 0}^{T\times T}$ of pair-wise distances, defined as

$$\psi(\mathbf{A}_{\text{m}}) := \text{M}_{t,t'} = \Big(\sum_{c=0}^{C-1}(|\text{A}_{\text{m}[c,t]} - \text{A}_{\text{m}[c,t']}|)^p\Big)^{1/p}, \quad (6)$$

for $p = 1$ and $t, t' \in [0, \ldots, T-1]$. Only for, and prior to, the computation of the $\mathbf{M}$, $\mathbf{A}_{\text{m}}$ is normalized so that the sum of the features at each time-frame $t$ sum up to unity. Furthermore, $\mathbf{P}_\lambda \in \mathbb{R}_{\geq 0}^{T\times T}$ is the transportation plan that is computed by solving the minimization problem

$$\mathbf{P}_\lambda = \underset{\mathbf{P}\in\mathbb{U}(r,c)}{\arg\min}\langle\mathbf{P}, \psi(\mathbf{A}_{\text{m}})\rangle - \frac{1}{\lambda}H(\mathbf{P}), \qquad (7)$$

where $H(\cdot)$ denotes the entropy function and $\lambda > 0$ is a scalar the controls the strength of the entropic regularization. $\mathbb{U}(r,c)$ is the set of non-negative matrices of size $T \times T$ whose rows and columns sum up to $r$ and $c$, respectively, where $r = c = 1$. For solving the minimization problem of Eq.(7) we employ the algorithm presented in (Cuturi, 2013) that is based on the Sinkhorn iterative matrix scaling operator (Sinkhorn, 1967).

## 2.2. Decoder

The decoder $D(\cdot)$ takes as an input the representation $\mathbf{A}_{\text{v}}$ and yields an approximation of the clean singing voice signal $\mathbf{x}_{\text{v}} \in \mathbb{R}^N$, denoted by $\hat{\mathbf{x}}_{\text{v}} \in \mathbb{R}^N$. Specifically, $D(\cdot)$ models the clean singing voice as a sum of $C$ modulated sinusoidal components that overlap in $\mathbb{R}^N$. The components are computed using an 1D transposed convolutions with $S$ strides and another set of $C$ number of kernels, $\mathbf{w}_c \in \mathbb{R}^L$, as

$$\hat{\text{x}}_{\text{v}[St+l]} = \eta + \sum_{c=0}^{C-1}\text{A}_{\text{v}[c,t]}\text{w}_{c[l]}, \text{ where} \qquad (8)$$

$$\eta = \begin{cases} 0, & \text{if } t = 0 \\ \hat{\text{x}}_{\text{v}[S(t-1)+l]}, & \text{otherwise} \end{cases}. \qquad (9)$$

As can be seen from Eq (9), $\eta$ is is a past sample contained in $\hat{\mathbf{x}}_{\text{v}}$, that is used for the overlap-add process. Regarding the kernels $\mathbf{w}_c$ of the decoder, in (Mimilakis et al., 2020) is proposed their re-parameterization as

$$\text{w}_{c[l]} = \cos(2\pi f_c^2 l + \rho_c)\,\text{b}_{c[l]}, \qquad (10)$$

where $\cos(\cdot)$ is the cosine function, and $l = [0, \ldots, L-1]$ is the time index. The parameters that are joinlty learnt with the parameters of the DAE are the sampling-rate-normalized carrier frequency $f_c$, the phase $\rho_c$ (in radians), and the modulating signal $\mathbf{b}_c \in \mathbb{R}^L$. The direct access to natural quantities like the above, significantly boosts the interpretability of the representation learning method. Additionally, $\mathbf{w}_c$ can be sorted according to the carrier frequency $f_c$, promoting intuitive representations.

After the reconstruction of $\hat{\mathbf{x}}_{\text{v}}$, the negative signal-to-noise ratio (neg-SNR) (Kavalerov et al., 2019), is computed as

$$\mathcal{L}_{\text{neg-SNR}}(\mathbf{x}_{\text{v}}, \hat{\mathbf{x}}_{\text{v}}) = -10\log_{10}\Big(\frac{||\mathbf{x}_{\text{v}}||_2^2}{||\mathbf{x}_{\text{v}} - \hat{\mathbf{x}}_{\text{v}}||_2^2}\Big), \qquad (11)$$

where $||\cdot||_2$ is the $\ell_2$ vector norm, and the negative sign is used to cast the logarithmic SNR as a minimization objective. Then, the overall overall minimization objective for $E(\cdot)$ and $D(\cdot)$ is computed using $\mathcal{L}_{\text{TV}}$ as

$$\mathcal{L}_A = \mathcal{L}_{\text{neg-SNR}} + \omega\,\mathcal{L}_{\text{TV}}, \qquad (12)$$

or using $\mathcal{L}_{\text{SK}}$ as

$$\mathcal{L}_B = \mathcal{L}_{\text{neg-SNR}} + \omega\mathcal{L}_{\text{SK}}, \qquad (13)$$

where $\omega$ is a scalar that weights the impact of the representation objective (either $\mathcal{L}_{\mathrm{TV}}$ or $\mathcal{L}_{\mathrm{SK}}$) in the learning signal for $E(\cdot)$.

## 3. Experimental Procedure

### 3.1. Dataset

For training and testing the representation learning method we use the freely available MUSDB18 dataset (Rafii et al., 2017). The dataset consists of 150 two-channel professionally produced multi-tracks, i.e, the stereophonic signals of bass, drums, singing voice, and other music instruments, that comprise a music mixture. Every signal is sampled at 44100 Hz. The multi-tracks are split into training (100 multi-tracks) and testing (50 multi-tracks) subsets.

### 3.2. Training

During training we sample a set of four multi-tracks from which we use the vocals and the other music instrument sources, collectively forming the accompaniment source. The accompaniment source is computed by adding the bass, drums, and other music instrument sources. Then, each sampled multi-track is down-mixed to a single channel and is partitioned into overlapping segments of $N = 44100$ samples. The overlap is 22050 samples. We randomly shuffle the segments for each source and corrupt the singing voice signal using the shuffled segments of the accompaniment source. For the corruption by additive Gaussian noise, the standard deviation of the noise is set to $1e - 4$.

For optimizing the parameters of the representation learning method, with respect to the minimization of Eq.(12) or Eq.(13), we use the adam algorithm (Kingma & Ba, 2015), with a batch of 8 segments and a learning rate of $1e - 4$. To compute the Sinkhorn distance(s), we average within the batch, all the cost matrices $\mathbf{M}$ computed using Eq.(6) and each $\mathbf{A}_{\mathrm{m}}$ contained in the batch.

### 3.3. Evaluation

For evaluating the usefulness of the representation that is learned by our method, we use the rest of the 50 tracks. Each track is down-mixed and partitioned into non-overlapping segments of $N = 44100$ samples (1 second length). Shuffling and random mixing is not performed at this stage. However, silent segments of the singing voice are discarded. The representation is evaluated with respect to the three following criteria: i) reconstruction error of the proposed method to encode and decode the clean singing voice signal using the previously described methodology, ii) reconstruction error of the separated singing voice signal by binary masking, and iii) additivity of the representation. The first two criteria are objectively measured with respect to the clean singing

voice signal $\mathbf{x}_{\mathrm{v}}$ using the scale-invariant signal-to-distortion ratio (SI-SDR) (Roux et al., 2019). Details regarding the computation of SI-SDR and the separation by binary masking are given in the supplementary material. Binary masking is used because it is an indicator of how disjoint (i.e. non-overlapping) two sources are, given a representation (more information exists in the supplementary material). We assess the additivity of the sources by computing the measure

$$\mathcal{A}(\mathbf{x}_{\mathrm{m}}, \mathbf{x}_{\mathrm{v}}, \mathbf{x}_{\mathrm{ac}}) = 1 - \frac{||E(\mathbf{x}_{\mathrm{m}}) - E(\mathbf{x}_{\mathrm{v}}) - E(\mathbf{x}_{\mathrm{ac}})||_1}{||E(\mathbf{x}_{\mathrm{m}})||_1 + \varepsilon}, \tag{14}$$

where $|| \cdot ||_1$ is the $L_1$ matrix norm, $\varepsilon = 1e - 24$ is a small term for ensuring numerical stability, and $\mathbf{x}_{\mathrm{ac}}$ is the time-domain signal of the accompaniment music source that is computed by mixing the multi-tracks available in the testing subset. High values of $\mathcal{A}(\cdot)$ indicate that the representation of the mixture signal consists of non-negative and additive sources (i.e. higher $\mathcal{A}(\cdot)$ is better). The attribute of additivity is important for the computation of optimal separation masks (Liutkus & Badeau, 2015), and in the unsupervised exploitation of music sources' structure (Smaragdis et al., 2006; Huang et al., 2012).

## 4. Results & Discussion

Table 1 contains the average and standard deviation values of the additivity measure $\mathcal{A}(\cdot)$, the SI-SDR for the reconstruction and the separation objective performance in dB, and the values of the hyper-parameters $\omega$ and $\lambda$. The results in Table 1 are discussed according to the SI-SDR value (higher is better), because SI-SDR is the reconstruction objective.

*Table 1.* Results from objectively evaluating the learned representations. Boldfaced values denote best obtained performance.

| Objective | $\omega$ | $\lambda$ | SI-SDR (dB) | SI-SDR-BM (dB) | $\mathcal{A}(\cdot)$ |
|---|---|---|---|---|---|
| $\mathcal{L}_A$ | 0.5 | N/A | 31.49 ($\pm$2.98) | 4.43 ($\pm$4.98) | 0.76 ($\pm$0.10) |
| | 1.0 | N/A | 31.39 ($\pm$3.16) | 4.66 ($\pm$4.92) | 0.76 ($\pm$0.10) |
| | 1.5 | N/A | 31.01 ($\pm$3.13) | 4.97 ($\pm$4.93) | 0.75 ($\pm$0.10) |
| | 2.0 | N/A | 30.96 ($\pm$2.98) | 4.65 ($\pm$4.90) | 0.76 ($\pm$0.10) |
| | 4.0 | N/A | 31.40 ($\pm$2.83) | 5.06 ($\pm$4.97) | 0.76 ($\pm$0.10) |
| $\mathcal{L}_B$ | 1.0 | 0.1 | 31.28($\pm$2.98) | 5.40($\pm$5.31) | 0.76($\pm$0.09) |
| | 1.0 | 0.5 | **31.61**($\pm$3.38) | **5.63**($\pm$**5.29**) | 0.77($\pm$0.09) |
| | 1.0 | 1.0 | 31.29($\pm$3.25) | 4.33 ($\pm$5.28) | 0.86($\pm$0.08) |
| | 1.0 | 1.5 | 29.98($\pm$3.48) | 0.06 ($\pm$6.43) | **0.89**($\pm$**0.08**) |
| | 1.0 | 2.0 | 31.13($\pm$3.66) | -0.02($\pm$6.44) | **0.89**($\pm$**0.08**) |

There are two observable trends in Table 1. The first trend is that when using $\mathcal{L}_B$, small values of $\lambda$ marginally improve the SI-SDR, compared to the best SI-SDR when using $\mathcal{L}_A$ (i.e. $\omega = 0.5$ and SI-SDR=31.49). Specifically, for $\lambda = 0.5$ and when using $\mathcal{L}_B$, we obtain an improvement of 0.12 dB and 1.20 dB for SI-SDR and SI-SDR-BM, respectively, compared to the case of using $\mathcal{L}_A$ and $\omega = 0.5$. Additionally, with the same $\lambda = 0.5$ for $\mathcal{L}_B$, we obtain an improvement of 0.57 dB SI-SDR-BM, compared to the best SI-SDR-BM with $\mathcal{L}_A$ (i.e. with $\omega = 4.0$). This trend shows

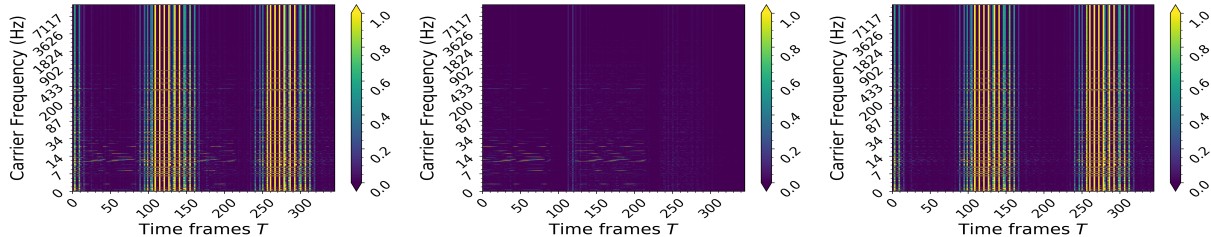

*Figure 2.* Learned representations for the mixture (left), the singing voice (middle), and the accompaniment (right) signals using the $E(\cdot)$ optimized with $\mathcal{L}_B$ for $\mathcal{L}_{\text{SK}} : \omega = 4.0, \lambda = 1.5$

that when using Sinkhorn distances as an objective (i.e. $\mathcal{L}_B$) and small entropic regularization weight (i.e. small values of $\lambda$), there is a marginal improvement of the reconstruction performance for the singing voice (measured with SI-SDR-BM), but also the learned representations yield better results for singing voice separation (measured with SI-SDR).

The second trend observed in Table 1 is that when using $\mathcal{L}_B$ and $\lambda > 1$, specifically for $\lambda \in [1.5, 2.0]$, the SI-SDR for binary masking drops by more than 5 dB, compared to $\lambda = 0.5$. This indicates that the separation by binary masking fails, suggesting that the singing voice and accompaniment are completely overlapping in the representation of the mixture $\mathbf{A}_{\text{m}}$. That is expected since entropy expresses the uncertainty about the representation of the mixture signal. This means that during training, all the elements in the feature space of the representation are equally probable to be active when the mixture signal is encoded. However, that uncertainty comes with an observed effect that is the sources become additive in the learned representation.

To further investigate the effect of entropic regularization with respect to the additivity metric, we keep the best $\lambda = 1.5$ from Table1, and examine the impact of the weight $\omega$ on $\mathcal{L}_B$. The corresponding results compared to the STFT, that is the most commonly employed representation for music source separation, are given in Table 2. The results from Table 2 suggest that by increasing the weight $\omega$ that affects the strength of the representation objective in the learning signal, the learned mixture representations, for $\omega = 4.0$, consist of two almost additive representations, i.e., the singing voice and the accompaniment representations. Furthermore, nearly all representations computed using the Sinkhorn distances and the entropic regularization, outperform the STFT with respect to the objective measure of additivity in an unsupervised fashion.

*Table 2.* Objective evaluation of the additivity of the learned representations.

| Objective | $\omega$ | $\lambda$ | $\mathcal{A}(\cdot)$ |
|---|---|---|---|
| | 1.0 | 1.5 | 0.89 ($\pm 0.08$) |
| $\mathcal{L}_B$ | 1.5 | 1.5 | 0.90 ($\pm 0.07$) |
| | 2.0 | 1.5 | 0.92 ($\pm 0.07$) |
| | 4.0 | 1.5 | **0.93** ($\pm$**0.06**) |
| STFT | N/A | N/A | 0.86 ($\pm 0.06$) |

To qualitatively assess the representations for the extreme case observed in Table 2, Fig. 2 illustrates learned representations for the mixture, singing voice, and the accompaniment signal. The signals were acquired from a single multi-track segment contained in the testing sub-set of MUSDB18. The representations are computed using the optimized encoder with the $\mathcal{L}_B$ objective. As it can be clearly observed from Fig. 2 higher than 0.5 entropic regularization enables the learning of representations that for particular sources such as the accompaniment, exhibit distinct structure, i.e., vertical activity (activity with respect to $C$). Furthermore, the representation of the singing voice is characterized by horizontal activity, i.e., a few components $C$ are active and smoothly vary in time. We believe that the distinct structure of the music sources, observed in Fig.2, could be useful for unsupervised separation and/or enhancement methods such as the deep audio prior (Michelashvili & Wolf, 2019) and the harmonic convolution(s) model (Zhang et al., 2020).

## 5. Conclusions

In this work we proposed the usage of entropic regularized Sinkhorn distances as a cost objective for unsupervised learning of interpretable music signal representations. We experimentally showed that Sinkhron distances can be useful for the problem of learning representations for singing voice separation. Particularly, the learned representations allow the separation of singing voice by masking for small values of entropic regularization, improving a previously proposed unsupervised approach. Nonetheless, higher values of entropic regularization lead to learning representations of sources that are distinctly structured and are almost additive; attributes that are useful in music source separation. The source code is based on the Pytorch framework (Paszke et al., 2019) and is freely available online[1].

## Acknowledgements

Stylianos I. Mimilakis is supported in part by the German Research Foundation (AB 675/2-1, MU 2686/11-1). K. Drossos would like to acknowledge CSC Finland for computational resources.

---

[1] https://github.com/Js-Mim/rl_singing_voice

# Supplementary Material

## Computation of Sinkhorn Distances

The entropy for the regularization of Eq.(7) is computed as

$$H(\mathbf{P}) = -\sum_{t,t'=0}^{T-1} \mathrm{P}_{[t,t']} \log(\mathrm{P}_{[t,t']})$$

For solving Eq.(7) with the Sinkhorn iterative matrix scaling algorithm and entropic regularization we used the Algorithm 1 presented in (Cuturi, 2013). We set the total number of iterations to $1e3$ per each batch, and the termination threshold to $1e-5$.

The normalization of $\mathbf{A}_\mathrm{m}$ prior to the computation of the Sinkhorn distances is based on:

$$\mathrm{A}^*_{\mathrm{m}[c,t]} = \frac{\mathrm{A}_{\mathrm{m}[c,t]}}{\sum_c (\mathrm{A}_{\mathrm{m}[c,t]} + \frac{1}{C})}$$

## Hyper-parameter Selection

### Convolutional Networks

For training, the total number of iterations throughout the whole training sub-set is set to 10. The selection is based on the experimental procedure presented in (Mimilakis et al., 2020), suggesting that any improvements towards the minimization of the overall cost function do not take place after the 10-th iteration.

The hyper-parameters for the convolution kernels are based on the best performing combination that has been previously presented in (Mimilakis et al., 2020) and are: number of kernels for the convolutional encoder $C' = C = 800$, stride size used in the first convolutional operator and the decoder $S = 256$, length of each kernel in the first convolution and in the decoder $L = 2048$, length of the second convolution $L' = 5$, and the dilation factor of the second convolution $\phi = 10$.

### Audio Signals & Transforms

In the evaluation and for the comparison with the STFT, the STFT uses a window size of 2048 samples, an analysis step-size of 256 samples and the Hamming windowing function. The window-size and the step-size were selected according to the closest match of the hyper-parameters in the convolutions (stride, and kernel length).

The removal of silent segments is based on the following:

$$l_{\mathrm{x_v}} = 10\log_{10}(||\mathbf{x}_\mathrm{v}||_2^2 + \epsilon) \begin{cases} \mathbf{x}_\mathrm{v} : \text{active}, & \text{if } l_{\mathrm{x_v}} \geq -10 \\ \mathbf{x}_\mathrm{v} : \text{silent}, & \text{otherwise}. \end{cases}$$

## Initialization

The kernels in the first convolutions are randomly initialized with values drawn from a uniform distribution. The bounds of the uniform distribution are $(-\sqrt{\frac{3}{C}}, \sqrt{\frac{3}{C}})$, following the initialization strategy presented in (He et al., 2015). For the decoder, the phase values $\rho_c$ are initialized to zero, and all the elements of the modulating vectors $\mathbf{b}_c$ are initialized to $\frac{1}{C+L}$. The initialization of the normalized frequencies $f_c$ is inspired by (Ravanelli & Bengio, 2018) and is performed by first computing the center frequencies of the Mel scale $f_\mathrm{Mel}$ between $f_\mathrm{Hz} \in [30, \ldots, 22050]$ Hz, over $C = 800$ number of steps, using

$$f_\mathrm{Mel} = 2595 \log_{10}(1 + \frac{f_\mathrm{Hz}}{700})$$

and then the initial $f_c$ value is computed as

$$f_c = \frac{700 \, 10^{f_\mathrm{Mel}/2595} - 1}{44100}$$

## Separation by Binary Masking

We conduct singing voice separation by masking because masking is an important operation in audio and music source separation, and has been extensively used by DL-based approaches and also representation learning (Tzinis et al., 2020). The focus is given on informed separation, i.e., masks are computed by an oracle method using the information for all the mixture's sources from the dataset. This is done in order to estimate the least-upper-bound performance of singing voice separation for a learned representation. This alleviates the biases on the prior information that music source separation approaches have. Examples of biases include the source's structure and existing neural architectures engineered for the STFT. Finally, binary masking is used because it is an indicator of how disjoint (less overlap) two sources are given a representation.

The binary mask is computed by encoding three signals. The first signal is the mixture $\mathbf{x}_\mathrm{m}$, the second signal is the accompaniment source $\mathbf{x}_\mathrm{ac}$, and the singing voice signal $\mathbf{x}_\mathrm{v}$. Using the trained encoder $E(\cdot)$ the representations $\mathbf{A}_\mathrm{m}$, $\mathbf{A}_\mathrm{ac}$, and $\mathbf{A}_\mathrm{v}$ are computed for $\mathbf{x}_\mathrm{m}$, $\mathbf{x}_\mathrm{ac}$, and $\mathbf{x}_\mathrm{v}$, respectively. The mask $\mathbf{G}_\mathrm{v} \in \mathbb{R}^{C \times T}$ is computed as

$$\mathbf{G}_\mathrm{v} = g(\mathbf{A}_\mathrm{v} \oslash \mathbf{A}_\mathrm{ac}),$$

where "$\oslash$" is the element-wise division and $g(\cdot)$ is defined as

$$g(\mathrm{x}) = \begin{cases} 1, & \text{if } \mathrm{x} \geq 0.5 \\ 0, & \text{otherwise} \end{cases}.$$

The approximation of the singing voice time-domain signal $\hat{\mathbf{x}}_\mathrm{v}$ using binary masking is computed using

$$\hat{\mathbf{x}}_\mathrm{v} = D(\mathbf{A}_\mathrm{m} \odot \mathbf{G}_\mathrm{v}),$$

where "$\odot$" is the element-wise (Hadamard) product.

**Computation of SI-SDR**

The scale-invariant signal-to-distortion ratio in dB is computed for each segment, as

$$\text{SI-SDR}(\mathbf{x}_v, \hat{\mathbf{x}}_v) = 10 \log_{10}\left(\frac{||\alpha \mathbf{x}_v||_2^2}{||\alpha \mathbf{x}_v - \hat{\mathbf{x}}_v||_2^2}\right), \text{ for}$$

$$\alpha = \frac{\tilde{\mathbf{x}}_v^T \mathbf{x}_v}{||\mathbf{x}_v||_2^2}. \qquad (15)$$

Higher SI-SDR values indicate better reconstruction or separation performance.

**Additional Results**

In Figure 3 we demonstrate additional results from the objective evaluation of the learned representations using $\mathcal{L}_B$ that consists of the Sinkhorn distances. Particularly, Figure 3 contains error plots for a greater range of entropic regularization weights $\lambda \in [0.1, 0.5, 1.0, 1.3, 1.5, 2.0, 5.0, 10.0]$ and for $\omega = 1.0$. In addition to this, we have included results for $p = 1$ and $p = 2$, where $p > 0$ is used in the computation of the cost matrix $\mathbf{M}$ used by the Sinkhorn distances.

From Figure 3 two main observations are highlighted. The first observation is that the computation of the cost matrix $\mathbf{M}$ for $p = 2$ leads to marginally sub-optimal results, for nearly all $\lambda$ values and metrics, compared to $p = 1$. Specifically, the reconstruction performance of $p = 1$ is outperforming $p = 2$ by 1 dB, on average across $\lambda$ values. Also, $p = 1$ outperforms by 0.6 dB, on average, $p = 2$ for the separation by masking performance. For the additivity metric, $p = 2$ marginally outperforms $p = 1$ for a negligible difference of $3e^{-3}$. For these reasons the main results of our work focus on $p = 1$.

The second observation is that for $\lambda > 2$ the observed separation performance dip and additivity performance peak in the area of $\lambda \in [1.3, 1.5, 2.0]$ disappears, and the examined method performs similarly to the values of low entropy, according to the examined metrics. This contradicts our expectations for the effect of entropic regularization. Our only explanation to this behavior is that for values $\lambda > 2$ the exponential function used in the computation of the Sinkhorn distances and applied initially to $\mathbf{M}$ yields saturated values that bias the overall minimization, in an unexpected way, that requires a closer inspection.

In similar vein, for the $\mathcal{L}_A$ that uses the total-variation denoising cost the full results complimenting Table 1 are illustrated in Figure 4.

To justify the selection of the particular $\lambda = 1.5$ hyperparameter for computing $\mathcal{L}_B$ in Table 2, Figure 6 illustrates the evaluation results for neighbouring $\lambda \in [1.3, 2.0]$ compared to $\lambda = 1.5$, where similar behavior is observed. As it can be observed from Figure 6 the performance of all the

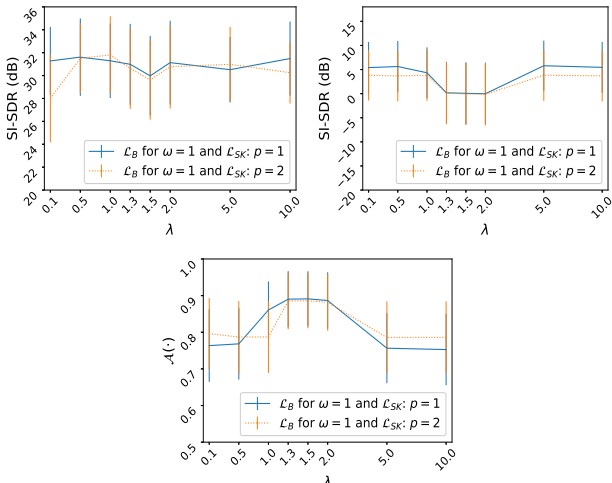

*Figure 3.* Performance evaluation of the learned representations by $\mathcal{L}_B$ that uses the Sinkhorn distances. (top-left) Reconstruction of singing voice in SI-SDR, (top-right) oracle separation performance in SI-SDR, and (bottom) additivity objective measure. Horizontal and vertical lines denote the average and the standard deviation of the performance, respectively.

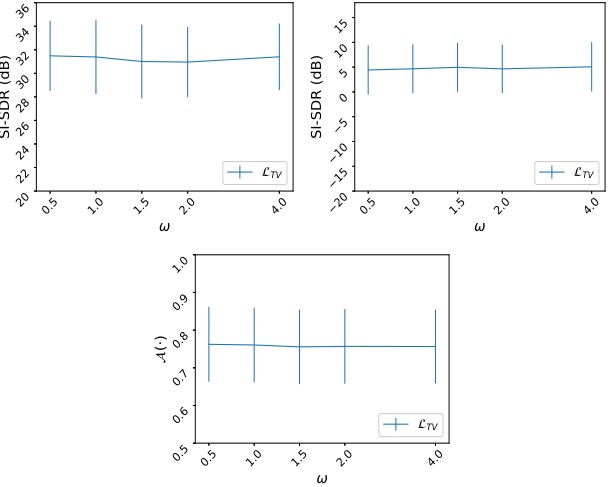

*Figure 4.* $\mathcal{L}_A$ using total variation denoising ($\mathcal{L}_{\text{TV}}$) for various values of $\omega$. Left reconstruction of singing voice in SI-SDR, middle oracle separation performance in SI-SDR, and right additivity objective measure

representations is nearly identical, with a negligible performance boost observed for of $\lambda = 1.5$ (orange line), on average across the values of $\omega$.

Finally, in Figure 5 we provide additional illustrations of the representations obtained using either $\mathcal{L}_A$ or $\mathcal{L}_B$, for a random multi-track segment. For $\mathcal{L}_B$ we focus on two extreme cases of separation and additivity performance observed from Tables 1 and 2. In particular, we illustrate representations obtained for entropy values $\lambda = 1.5$ and for $\lambda = 0.5$,

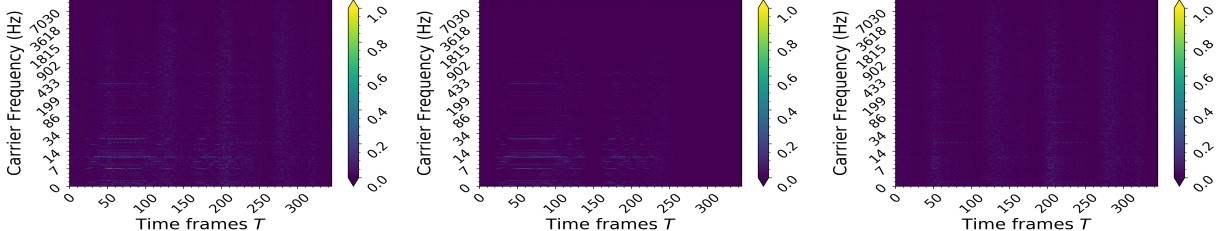

(a) Learned representations for the mixture (left), the singing voice (middle), and the accompaniment (right) signals using the $E(\cdot)$ optimized with $\mathcal{L}_A$ for: $\mathcal{L}_{\text{TV}} : \omega = 4.0$



(b) Learned representations for the mixture (left), the singing voice (middle), and the accompaniment (right) signals using the $E(\cdot)$ optimized with $\mathcal{L}_B$ for: $\mathcal{L}_{\text{SK}} : \omega = 1.0, \lambda = 0.5$

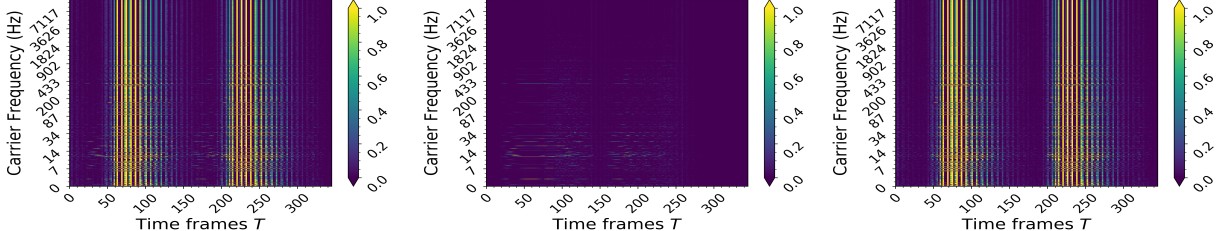

(c) Learned representations for the mixture (left), the singing voice (middle), and the accompaniment (right) signals using the $E(\cdot)$ optimized with $\mathcal{L}_B$ for $\mathcal{L}_{\text{SK}} : \omega = 4.0, \lambda = 1.5$

*Figure 5.* An illustration of the learned representations of a single multi-track segment, using three optimized encoders $\mathcal{E}$.

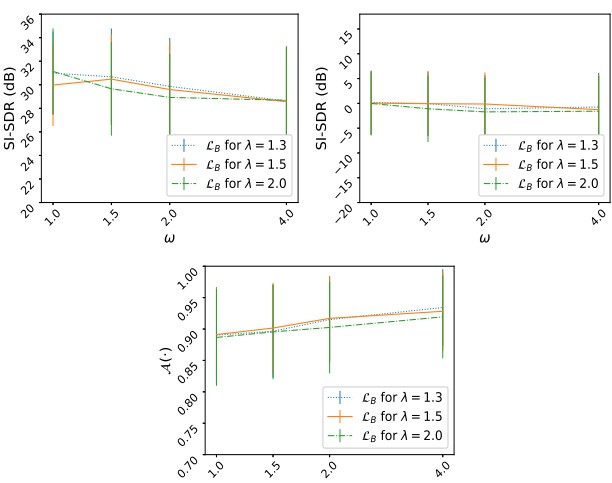

*Figure 6.* Performance evaluation of the learned representations by $\mathcal{L}_B$ using three entropic regularization $\lambda$. (top-left) Reconstruction of singing voice in SI-SDR, (top-right) oracle separation performance in SI-SDR, and (bottom) additivity objective measure. Horizontal and vertical lines denote the average and the standard deviation of the performance, respectively.

that resulted in the best performance of additivity and masking, respectively. For comparison, we also display learned representations for $\mathcal{L}_A$ for $\omega = 4.0$, in which the best separation performance for $\mathcal{L}_A$ was observed in Table 1.

By observing Figure 5 it can be seen that the usage of $\mathcal{L}_A$ (employing the total-variation denoising cost) leads to smooth representations. However, qualitatively the representation of the mixture and the sources seem somewhat blurry, without distinct structure between the sources. Consequently, representations learned using $\mathcal{L}_A$ might impose difficulties for source separation methods. On the other hand the employment of $\mathcal{L}_B$ with the Sinkhorn distances and for $\lambda = 0.5$, leads to learned representations that at least for the singing voice signal a prominent structure of horizontal activity is observed. The interesting part comes when the entropy regularization weight is increase to $\lambda = 1.5$, where now the accompaniment source is distinguished by prominent vertical activity.

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
