# OpenReview forum: "Revisiting Representation Learning for Singing Voice Separation with Sinkhorn Distances"
_ICML.cc/2020/Workshop/SAS — Submitted to SAS 2020_

### Official Review · AnonReviewer1 · 2020-06-26

**Rating:** 4
**Confidence:** 3

**Review:**

Authors presented a strategy based on denoising autoencoders aimed at separating singing voice from music. While the encoder learns spectral-like representations, the decoder maps it back to the time-domain, filtering away the non-vocal components. The idea is natural and interesting and I appreciate the fact that authors work directly on the time domain, however I highlight some concerns below, along with questions and suggestions to the authors.

While the authors claim the method is unsupervised, the ground truth singing voice signal is required at training time as a supervision signal. As I see, an unsupervised approach would learn to isolate voice from mixture signals directly. While I leave the analysis as to whether the paper fits in the scope of the workshop to the organizers, I strongly recommend the authors to rephrase the manuscript in order to highlight the method is actually supervised (which doesn’t reduce its relevance in my opinion).

It is also unclear to me what is the research question being addressed in the paper. Perhaps state it clearly in the introduction.

It is unclear to me how the proposed model behaves at test time once only a single input signal is available. It would be important to explain that in the text or perhaps include a similar diagram as that in Fig. 1 but for the inference case, given that a single audio signal is available at test time, i.e. once only x_m is available.

I further have some concerns/suggestions regarding specifically the evaluation, the main one being the fact that the method’s performance is not compared with any alternative approach (no baselines). More details are highlighted below:

Ablations - Since all conclusions are drawn from empirical analysis, I believe some ablation experiments are required. I specifically recommend the following: (i) Remove the encoder and evaluate similar decoders reconstructing directly from power spectra, and (ii) Remove the auxiliary loss and train the same encoder and decoder with only the reconstruction loss. Report reconstruction results in both cases.

Training hyperparameters selection: more training details should be added to enable reproduction. Was there tunning for each reported model or a shared set of training hyperparameters? Empirical evidence would be stronger if at least a small grid were used in each case (each line in each table should be the best in the grid) and such grid should be reported.

It would be interesting to listen to some samples separated by the model (both the models already reported and suggested ablations and other baselines for comparison).

---

### Official Review · AnonReviewer2 · 2020-06-29

**Rating:** 6
**Confidence:** 3

**Review:**

The authors extend a previous work based on denoising-autoencoders to a new specific loss with the overall goal to separate singing voices from music. While the idea is only incremental with respect to previous works, results are interesting. More precisely, it is important to explore models that are able to directly deal with time-domain signals to facilitate the global interpretation. Auto encoders are a very nice and simple way to learn new representations. First clean voice samples are corrupted by instrumental tracks and/or gaussian noise before being fed to the encoder that learns a "spectral" representation of the signal. Then, the decoder maps this representation back to the time-domain based on the clean singing sample.

Note that the term: "unsupervised" in this application may be subject to debate. Indeed, the task is to obtain clean singing voices, and the authors also need clean singing voices to train the model.

The first problem I have is directed toward the additive noises (or instruments) added to the clean voice sample. It is most likely that a system trained on synthetic data will end up with poor performances on the wild due to the controlled and restricted distributions of perturbations that it is able to deal with. Thus, I would have liked to see a higher diversity on the evaluation protocol with the use of a second dataset.

The second concern I have is linked to the lack of comparison with other methods. Here, we are mostly interested to see if the Sinkhorn distance is better than a more traditional approach but we also need a baseline to compare with, in order to better understand the empirical impact of this modification.

Could be great to release some generated samples to listen to!

+ Pros +

+ Well-written.
+ Interesting approach.
+ Results seems to be good.

- Cons -

- The introduction lack a bit of motivation to grasp the interest of such a research domain and trajectory.
- No comparison with other methods.
- Unsupervised in the data, but not unsupervised in the training procedure w.r.t the task
- Scaling to realistic tasks due to synthetic perturbations used during training.

---

### Official Review · AnonReviewer3 · 2020-06-30
**Unsupervised approach for singing voice separation with Sinkhorn distance as regularizer**

**Rating:** 6
**Confidence:** 4

**Review:**

The paper builds on prior work by the authors in recent conferences, notably in Eusipco 2020 where they proposed the core algorithm   which the paper extends.  The proposed approach builds upon the denoising auto encoder model and uses sinusoids to reconstruct the original signal in the decoder. In prior work the authors used a smoothness enforcing loss function which looked at mean square error between shifted representations in both the rows and columns of the encoded matrix representation of the time domain waveform. In this paper they explore a sinkhorn distance based smoothing function based on recent work which argues that for low dimensional manifolds, the sinkhorn distance is better.
My concerns with this paper are that the change in loss function for regularization and smoothing is not well motivated beyond referring to the wasserstein GAN paper (Arjovsky 2017). There are simpler techniques possible in extending the MSE idea itself such as considering multiple shifts or variations to the L2 norm. Also while the results indicate  that the proposed loss function improves over the MSE approach the evaluation was done on one corpus and one task which make it hard to comfortably generalize the conclusions highlighted in the paper.

The paper is otherwise well written though the authors have moved a lot of relevant material on evaluation and metrics into the additional material section which make it hard to read and understand within the confines of the core paper. The experimental qualitative discussion around Figure 2 is hard to follow. It is a struggle to understand if the middle subfigure has any content at all. Another important piece of information missing is the range of hyperparameters tried for the model architecture itself and whether that has a bearing on the results

---

### Decision · Program_Chairs · 2020-07-01

**Decision:**

Reject

**Comment:**

Dear author(s),

Thank you very much for your submission at the ICML2020@SaS workshop (https://icml-sas.gitlab.io/). Based on the scores assigned by the reviewers, we regret to inform you that the paper was rejected. We got 26 submissions and we were only able to accept 13 papers. We invite you anyway to consider the feedback of the reviewers and to follow our upcoming workshop on July 17.